# New measures of agency from an adaptive sensorimotor task

**Shiyun Wang**[1], **Sivananda Rajananda**[1], **Hakwan Lau**[1,2,3,4], **J. D. Knotts**[1]*

**1** Department of Psychology, University of California, Los Angeles, California, United States of America,
**2** Brain Research Institute, University of California, Los Angeles, California, United States of America,
**3** Department of Psychology, University of Hong Kong, Pok Fu Lam, Hong Kong, **4** State Key Laboratory of
Brain and Cognitive Science, University of Hong Kong, Pok Fu Lam, Hong Kong

* jeffreydknotts@gmail.com

England, AUSTRALIA

**Data Availability Statement:** Code and data files
for the experiment are available from the Kaggle
database (URL: www.kaggle.com/dataset/
a2da3634c9e4971fcc031e544c8019a4623a9998
658363c7bf9808c07e1b43d8)

## Abstract

Self-agency, the sense that one is the author or owner of one's behaviors, is impaired in multiple psychological and neurological disorders, including functional movement disorders, Parkinson's Disease, alien hand syndrome, schizophrenia, and dystonia. Existing assessments of self-agency, many of which focus on agency of movement, can be prohibitively time-consuming and often yield ambiguous results. Here, we introduce a short online motion tracking task that quantifies movement agency through both first-order perceptual and second-order metacognitive judgments. The task assesses the degree to which a participant can distinguish between a motion stimulus whose trajectory is influenced by the participant's cursor movements and a motion stimulus whose trajectory is random. We demonstrate the task's reliability in healthy participants and discuss how its efficiency, reliability, and ease of online implementation make it a promising new tool for both diagnosing and understanding disorders of agency.

## Introduction

Self-agency refers to the feeling that one is the cause of their behaviors [1]. Impairments in self-agency have been observed in multiple psychological and neurological disorders, such as functional movement disorders (FMDs) [2], Parkinson's disease [3], alien hand syndrome [4], schizophrenia [5], and dystonia [6]. However, existing measurements of agency are rarely applied in diagnosis due to concerns of unreliability [7–9]. Behavioral tasks in general have been shown to have low test-retest reliability as measures of individual differences and predictors of real-life outcomes [10,11], suggesting limited efficacy in diagnostic procedures. Moreover, prominent behavioral measurements of agency contain their own specific limitations. For example, the Libet clock paradigm is widely used to assess movement-based self-agency in experimental settings [12]. In this task, a dot loops around a clock face with a uniform speed. While tracking the movement of the dot, participants can press a button to stop the dot with a variable delay. Depending on the instruction, they then report the location of the dot at the timing of movement intention initiation or movement execution. The Libet task has been criticized for confounding perceived timing of voluntary movement initiation and execution with

**Funding:** The author(s) received no specific funding for this work.

**Competing interests:** The authors have declared that no competing interests exist.

timing for memory processing [13], and for its general susceptibility to memory-based biases [14–16]. Taken together, this suggests that a reliable and unbiased behavioral method for assessing agency could represent a major step forward in the diagnosis of agency related disorders [17].

Among the disorders mentioned above, we focus on FMDs here, as their diagnosis is particularly challenging. FMDs are movement disorders characterized by body movements or postures that patients cannot control, and that have no known neurological basis [18]. The most prominent feature of FMDs is patients' lack of a sense of agency over body movements or postures [2]. When this happens, patients feel that they do not own the behaviors their bodies carry out. In other words, their bodies exert random movements, such as tremor and twitches, that do not feel self-initiated and cannot be controlled. This is often reported to be disruptive and, in some cases, can lead to a prolonged state of disability [19]. A sense of agency is acquired if little or no discrepancy is observed between movement intention and sensory feedback [20]. However, patients with FMDs report impaired movement intention, which leads to a frequent mismatch between intention and feedback. For example—setting concerns about the reliability of the Libet clock paradigm to the side temporarily—in contrast to healthy controls, FMD patients show no difference in reported intention and movement times on the Libet clock paradigm [12,21]. Further, the right temporo-parietal junction (rTPJ) is thought to contribute to the generation of self-agency by comparing the prediction of movements with actual sensory feedback [22,23]. Decreased connectivity between the rTPJ and sensorimotor regions has been observed in FMD patients, therefore suggesting that an impairment in the ability to generate a sense of movement agency may underlie symptoms [24].

Current diagnosis of FMDs is ambiguous and time-consuming [25]. One challenge with FMD diagnosis is that about 25% of patients with FMDs have other neurological illnesses, including separate movement disorders with known neurological causes [19]. These separate movement disorders may hinder detection of FMDs given their similarity in symptoms. It is also difficult to determine whether certain psychological factors, such as anxiety, are causes or consequences of FMDs [25].

Though the Fahn and Williams clinical classification of FMDs is widely used [25–27], diagnoses require a large time investment. According to this procedure, an FMD patient's disorder should be "inconsistent over time or incongruent with a recognized [movement disorder], in association with other related features" [28,29]. Confirmation of this criterion requires extended observation and a clinician's thorough understanding of all prominent types of movement disorders. But because the early diagnosis of FMDs can benefit recovery [25], procedures that can increase the efficiency of diagnosis are very much in need.

One promising option in this regard is behavioral paradigms that assess agency by modulating the extent to which participants' can control moving stimuli with cursor movements [30,31]. For example, in one experiment, Wen et al., 2018 [30] had participants identify a target circle among a group of distractors, while the trajectories of the two types of stimuli were influenced by the participants' mouse movements to varying degrees. Such tasks involve more extensive sensorimotor interaction from participants, and do not rely on the types of discrete timing judgments, like those used in the Libet clock paradigm [12], that have been criticized for susceptibility to memory confounds. In the current study, we took a similar approach and developed an adaptive online task in which participants use cursor movements to affect the trajectory of one of two moving dots, and then make a forced-choice judgment about which dot was under their control. We show that this easily-implementable task can provide fast, reliable estimates of agency in healthy participants, and discuss its potential to improve diagnostic efficacy not only for FMDs, but for agency-related disorders in general.

## Methods

### Participants

We conducted two experiments with minor procedural differences through Amazon Mechanical Turk (mTurk). In Experiment 1, the data of 99 participants (71 males, average age = 29.66) was collected, and 53 participants were included in the data analysis (23 males, average age = 35.40; see Exclusion Criteria). In Experiment 2, the data of 94 participants (49 males, average age = 36.89) was collected, and 54 participants (28 males, average age = 37.24) were included in the data analysis.

Participants received $4 for finishing the task. People who partially completed the task were paid at a rate of $4 per hour. An online consent form was given at the beginning of the study. The study was approved by the University of California, Los Angeles North General Institutional Review Board (IRB number: 15–001476), and was carried out following the Declaration of Helsinki.

### Stimuli and apparatus

On each trial, two moving dots (dot A and dot B) were presented within separate circles for 4 or 2.5 seconds, in Experiment 1 and Experiment 2, respectively (Fig 1A and S1 Video). Each dot had an independent, pseudorandom trajectory, meaning that without the influence of cursor movements, the two dots will have different random trajectories. While the dots were

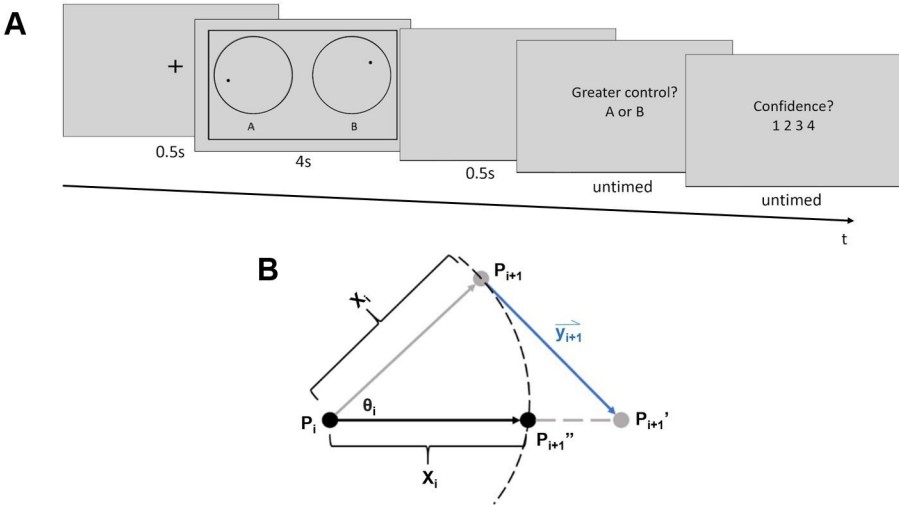

**Fig 1. Task procedure and calculation of dot trajectories.** A) Task procedure. At the beginning of each trial, a fixation cross was presented for 0.5 s, followed by the dot stimuli. When the dot stimuli were on the screen, participants could influence the moving trajectory of one of the dots with cursor movements. Then, after a blank screen (0.5 s), participants judged which of the dots they were better able to control (A or B), and how confident they were in this judgment on a scale from one to four. Font sizes are enlarged relative to their actual size during the experiment for clarity. See S1 Video for an illustration of example trials with mouse movements side-by-side with the dot motion stimuli. B) Calculation of dot trajectories. $X_i$ represents the distance of movement, which was randomly selected from a uniform distribution from 0.5 to 2.5 pixels on each new frame. An angle of movement ($\theta_i$) from frame i to i+1 was also generated for each dot at the beginning of each frame. It was randomly selected from a uniform distribution with a mean corresponding to the dot's angle of movement in the previous frame ($\theta_{i-1}$) and a range corresponding to that mean ± 11.75 degrees. $P_i$ represents the dot's location at the end of frame i-1. The dot was first moved along the angle $\theta_i$ by the distance $X_i$, to point $P_{i+1}$, which corresponds to the non-target dot's final location. Vector $y_{i+1}$ represents the mouse movement recorded during frame i-1 multiplied by the subject's current control level, and was used to shift the target dot to position $P_{i+1}$'. The target dot's final position, $P_{i+1}$", was then computed by finding the point along the line $P_iP_{i+1}$" that was $X_i$ away from the dot's original location ($P_i$).

presented, the participant could move the cursor to influence the trajectory of one of the two dots, hereafter referred to as the target dot. The instruction participants received regarding cursor movements stated: "On each trial, you will see two independent moving dots. The movement of your mouse can influence the trajectory of one of them . . . Your goal is to report which dot you are better able to control and how confident you are on your decision." No specific strategy for mouse movement was provided to avoid potentially biasing participants (e.g., toward shaking the cursor rapidly from side to side). The target dot (A or B) was determined randomly at the beginning of each trial.

Independent of cursor movements, random dot trajectories were computed as follows. The position of each dot was updated on each display frame. At onset, or frame i = 1, each dot appeared at the center of its respective circle. The initial angle of motion for each dot, $\theta_{i = 1}$ (Fig 1B), was randomly selected from a full 360-degree range. For successive display frames (i = 2 to i = n-1, where n is the total number of frames over which the dot stimuli were displayed) new angles of motion $\theta_i$ were randomly selected from a uniform distribution with a range of $\theta_{i-1} \pm 11.75$ degrees. Movement distance from frames i to i+1, $X_i$, (Fig 1B) was randomly selected from a uniform distribution from 0.5 to 2.5 pixels. Given a reference position, P, on display frame i, (Fig 1B, point $P_i$), the dot's position on frame i+1(Fig 1B, point $P_{i+1}$), was the dot's location after it moved along angle $\theta_i$ at a distance of $X_i$.

Cursor movements affected the trajectory of the target dot as follows. Cursor movement at frame i+1 was represented by a vector, $y_{i+1}$ (Fig 1B), that pointed from the cursor's coordinates at the time point when the coordinates of point $P_i$ were calculated (but before was $P_i$ drawn), to the cursor's coordinates at the onset of frame i+1. The amount of control that cursor movements had over the target dot's trajectory varied from trial to trial. To implement this, the cursor movement vector's length was multiplied by the participant's level of control (between 0 and 100%) on a given trial.

The position of the target dot on frame i+1 was computed by first moving the dot from position $P_{i+1}$ along the adjusted mouse movement vector to a new position, $P_{i+1}'$ (Fig 1B). The target dot's final display position on frame i+1, $P_{i+1}''$, was computed by moving the dot toward point $P_{i+1}'$ along the straight line connecting points $P_i$ and $P_{i+1}'$ by a distance of $X_i$.

If the non-target dot's computed position at frame i+1 ($P_{i+1}$, Fig 1B) was outside its circular border (Fig 1A), its coordinates were re-calculated so as to make it appear to "bounce" off of the border. The new position was re-calculated by reflecting the coordinates over the tangent line of the circular border at the point where the dot was in the previous frame ($P_i$). For the target dot, if its final position ($P_{i+1}''$) is outside the border, its final coordinate will adopt the value of its old coordinates ($P_i$). This was intended to minimize the extent to which the direction of cursor movements opposed the motion of the target dot.

Both dots were black and had a radius of 8 pixels. The cursor was hidden during the task to prevent participants from inferring the location of the target dot from cursor movement. If the cursor reached the edge of the screen, the corresponding side of an outer rectangular border (Fig 1A) changed color from black to red to alert participants that they had reached an edge of the area in which their cursor movements could be recorded.

Both experiments were conducted through mTurk. The experimental code was written in JavaScript and JsPsych (6.0.5). Participants performed the task with their personal computers.

## Procedure

On each trial in Experiment 1, a fixation cross was presented at the center of the screen for 0.5 seconds. Dot stimuli were then shown for 4 seconds. These were followed by a 0.5 s blank

screen, after which participants answered two questions. They first indicated which dot, A or B, they were better able to control. Then, they reported their confidence in this agency judgment on a scale of one to four. A rating of one corresponded to a complete guess. A two meant that the judgment was better than a guess but the participant was unsure about it. A three meant that they were almost certain, and a four meant that they had no doubt in their judgment. Participants had unlimited time to respond.

Each session started with five practice trials that implemented an adaptive staircasing procedure [32]. The control level of the first practice trial was 25%, as pilot data confirmed that healthy subjects have 100% accuracy at this level. If the participant's answer on a given trial was correct, the control level was reduced by 2.5% for the next trial. If the answer was incorrect, the control level was increased by 7.5% for the next trial. Participants would repeat the whole set of practice trials if their total accuracy was below 80% correct.

Following the practice, the main task employed two randomly interleaved one-up/one-down adaptive staircases with differentially weighted step sizes [32]. The ratio of down- to up-step magnitudes following correct and incorrect responses, respectively, was 0.33. This procedure was designed to estimate the control level at which participants are 75% correct in their agency judgments.

The control level on the first trial of each staircase was 2.5% and the initial down step size was 0.5%. Reversal trials were trials in which accuracy was different from that of the previous trial of the current staircase. For each staircase, the down step size was reduced to 0.25% after the second reversal, to 0.1% after the sixth reversal, and to 0.02% after the twelfth reversal. The experiment ended after both staircases accumulated 13 reversals. The maximum number of trials allowed was 100 trials in each staircase.

Experiment 2 was the same as Experiment 1 except for the following changes. To improve task efficiency, dot stimulus duration was decreased to 2.5 seconds, and participants only had 2 seconds to answer each question. Additionally, participants were given an optional break of up to 5 minutes in the middle of the task (after they had finished six reversal trials in both staircases). Finally, the last down step size was increased to 0.07% to prevent the staircasing procedure from becoming prematurely constrained to an overly narrow range of control values.

## Data analysis

Percent control thresholds on the dot trajectory task were estimated by computing the average control level across the last five reversal trials in each staircase. The final control threshold estimation for each participant was the average of the two staircases' thresholds.

Participants' metacognitive sensitivity was quantified using the area under the Type 2 receiver operating characteristic curve (Type 2 AUROC) [33]. According to Signal Detection Theory, participants' tasks in the current paradigm can be divided into two types: identifying the target dot is considered a Type 1 task and reporting confidence is considered a Type 2 task. The Type 2 AUROC reflects subjects' ability to track their performance on the Type 1 task with confidence ratings [33]. A Type 2 AUROC of 0.5 indicates that the participant has no metacognitive sensitivity, while a Type 2 AUROC of 1 indicates optimal metacognitive sensitivity (i.e., all correct responses are endorsed with high confidence while all incorrect responses are rated with low confidence).

Comparisons of control thresholds and Type 2 AUROC between the two experiments were made using two-tailed independent sample t-tests. An alpha level of .05 was used for all statistical tests. Statistical analyses were conducted in MATLAB R2018b (Natick, MA) and R version 3.6.0 (Vienna, Austria).

### Exclusion criteria

Catch trials in which participants had a high level of control (25%) were randomly inserted in between staircasing trials such that they accounted for approximately 15% of the total trial number. In Experiment 1, participants who missed more than 40% of catch trials were excluded (N = 39). Participants who used extreme confidence ratings (one or four) more than 95% of the time (N = 3) were also excluded because such biases would impede a meaningful analysis of metacognitive scores [34]. No participant was excluded due to a control level threshold that was more than three standard deviations away from the mean threshold level across participants. We also excluded participants whose d' scores [34,35] on the agency judgment task (N = 4), computed with hits corresponding to correctly choosing dot A and correct rejections corresponding to correctly choosing dot B, were less than or equal to zero, as such scores imply a lack of effort.

Similar exclusion criteria were applied to participants in Experiment 2. Twenty-eight participants were excluded for missing more than 40% of the catch trials. Four participants were excluded due to extreme confidence rating. No participants were excluded because of abnormal control level thresholds. Finally, eight participants were excluded for negative d' scores.

## Results

On average, participants finished the task in about 20 minutes (Experiment 1: M = 17.36 mins, SD = 9.34 mins; Experiment 2: M = 22.94, SD = 10.93). The distributions of percent control thresholds estimated in healthy participants in Experiments 1 (M = 1.90%, SD = 2.31%) and 2 (M = 1.47%, SD = 1.77%) are shown in Fig 2A and 2B, respectively. The reduction in stimulus duration and allotted response times from Experiment 1 to Experiment 2 did not significantly change observed control thresholds, t(97.52) = 1.07, p = 0.29.

Previous studies suggest that, compared to in-person subjects, online participants are less attentive to tasks, potentially resulting in poor data quality [36, but see also 37]. To control for this, we excluded participants based on their attentiveness, which was measured by their accuracy on the catch trials. Participants who missed more than 40% of the catch trials (Experiment 1: M = 7.67%, SD = 2.07%; Experiment 2: M = 6.97%, SD = 2.40%) had significantly higher control thresholds than participants who meet the catch trial criteria (Experiment 1: M = 2.55%, SD = 3.09%, t(94.332) = 9.77, p < .0001; Experiment 2: M = 2.21%, SD = 2.62%, t(55.469) = 8.57, p < .0001). Meanwhile, the thresholds of participants who were excluded by the catch trial criteria were significantly lower than the control level of catch trials (25%) in both experiments (Experiment 1: t(36) = -51.03, p < .0001; Experiment 2: t(27) = -39.80, p < .0001). These analyses imply that the excluded subjects were capable of performing well on the catch trials but failed to do so potentially due to their lack of attention to the task.

The trial counts in these experiments (Experiment 1: M = 83.64, SD = 21.93; Experiment 2: M = 83.61, SD = 22.25) were lower than that recommended for the unbiased metacognitive sensitivity estimation procedure used to estimate meta-d' [34]. Therefore, the method of Type 2 sensitivity estimation used here (Type 2 AUROC) is susceptible to being biased by Type 1 performance. Because the control levels at the start of each thresholding procedure were high by design, participants' Type 1 performance before the third reversal trial was relatively inflated. Thus, we computed Type 2 AUROC based on participants' performance after the third reversal trial in order to minimize the extent to which any initial inflation of Type 1 accuracy would bias the estimation of metacognitive sensitivity. Type 2 AUROC scores for Experiments 1 (M = 0.71, SD = 0.09) and 2 (M = 0.72, SD = 0.08) are shown in Fig 2C and 2D, respectively. No significant difference was observed between the two experiments, t(103.15) = -0.36, p = 0.72.

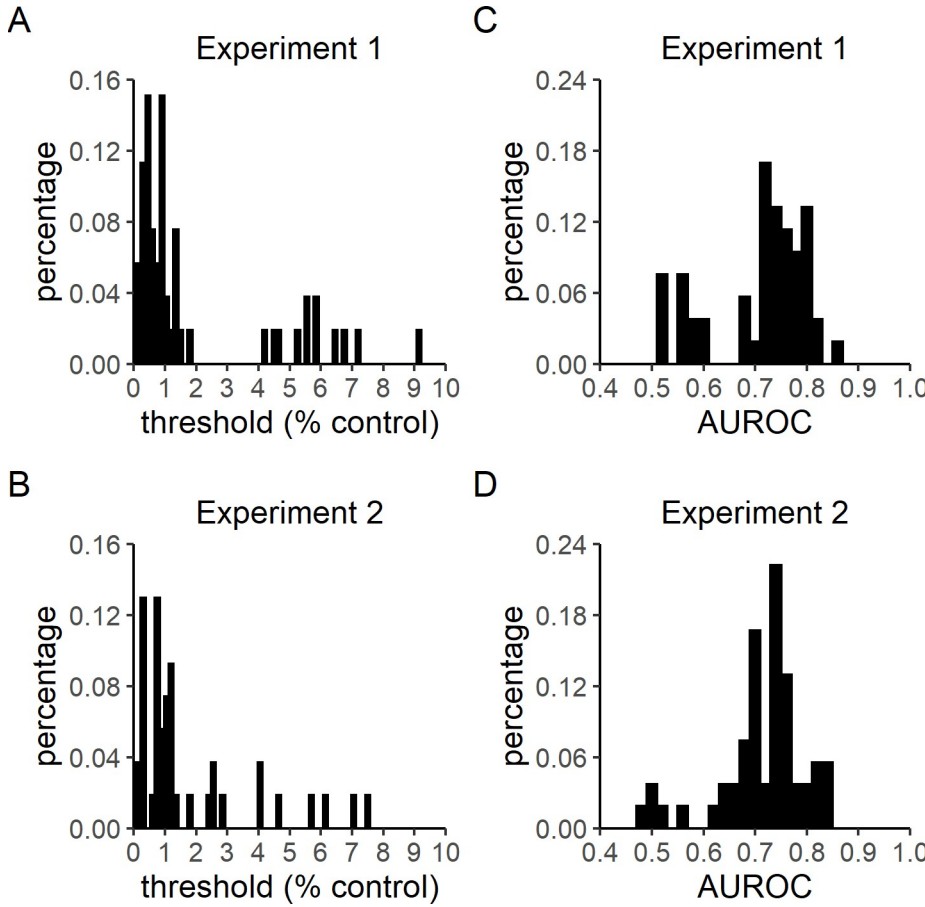

**Fig 2. Control threshold and Type 2 AUROC distributions.** Experiments 1 and 2 followed the trial procedure summarized in Fig 1, and had only slight differences in dot stimulus presentation time (4 versus 2.5 seconds, respectively), allotted response times (untimed versus 2 seconds, respectively), final staircasing step size (0.02% versus 0.07% control, respectively), and the availability of a midway break (Experiment 2 only). A) Control level thresholds in Experiment 1 (M = 1.90%, SD = 2.31%). B) Control level thresholds in Experiment 2 (M = 1.47%, SD = 1.77%). C) Type 2 AUROC in Experiment 1 (M = 0.71, SD = 0.09). D) Type 2 AUROC in Experiment 2 (M = 0.72, SD = 0.08). Bin sizes for threshold and AUROC histograms are 0.15 and 0.02, respectively.

Test-retest reliability of control level threshold estimation was assessed by evaluating the correlation between the average of each participant's control levels on the 8th to 10th reversal trials and the average of the subject's control levels on the 11th to 13th reversal trials. The 8th to 13th reversal trials were selected because, based on pilot data, staircases typically reached stable values starting at the 8th reversal trial. We aimed to verify the within-session stability of the staircasing procedure by performing the correlational test above. Spearman rank correlation tests were used to minimize the influence of extreme values. Positive correlations were observed in both experiments [Experiment 1: $r(51) = 0.90$, $p < 0.001$, Fig 3A; Experiment 2: $r(52) = 0.83$, $p < 0.001$, Fig 3B]. Because control thresholds were densely clustered at low levels, as shown by Shapiro-Wilk's tests [Experiment 1, reversals 8–10: $W(52) = 0.75$, $p < 0.001$; Experiment 1, reversals 11–13: $W(52) = 0.69$, $p < 0.001$; Experiment 2, reversals 8–10: $W(53) = 0.69$, $p < 0.0001$; Experiment 2, reversals 11–13: $W(53) = 0.66$, $p < 0.0001$; Fig 3A and 3B], split-half relationships between log transformed control thresholds are shown (Fig 4) to visually confirm that the observed reliability was not simply driven by extreme values.

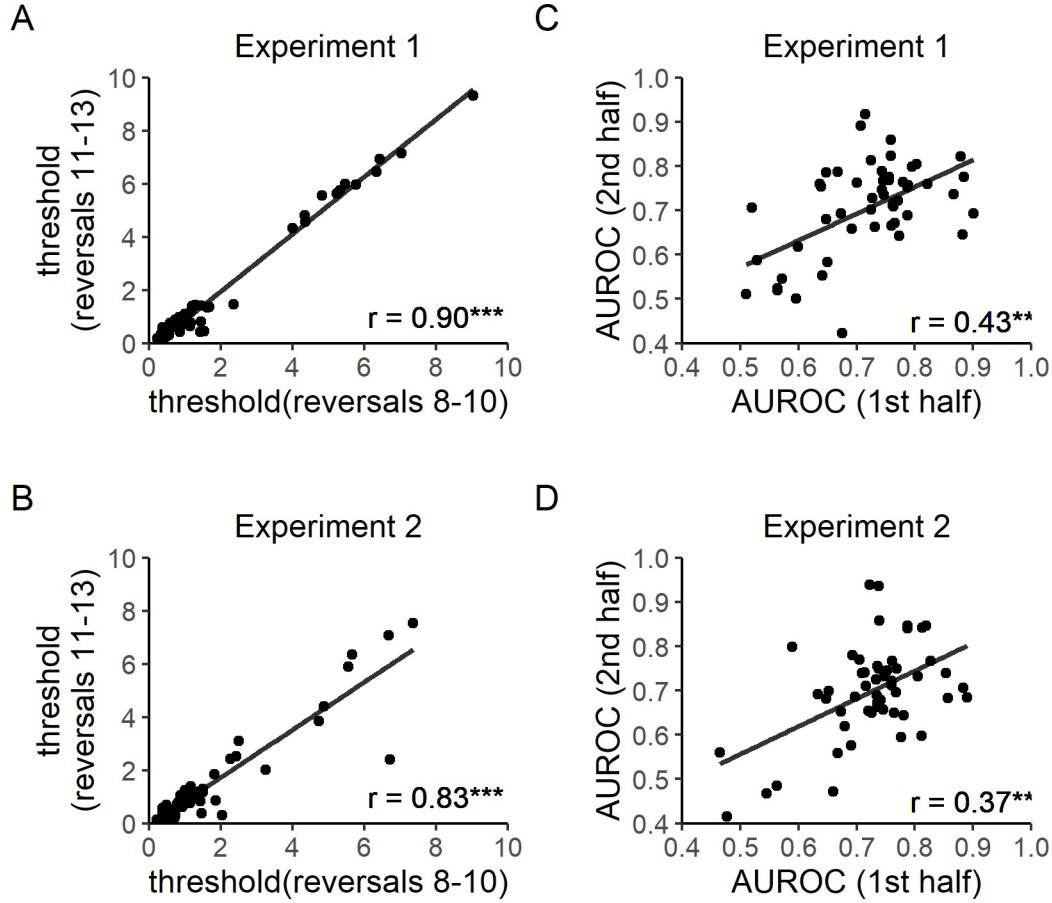

**Fig 3. Test-retest reliability.** A) Test-retest reliability of control level thresholds in Experiment 1. The average of each participant's control levels in the 8th to 10th reversal trials was correlated with that in the 11th to 13th reversal trials. A positive correlation was observed, r(51) = 0.90, p < 0.001. B) Test-retest reliability of control level thresholds in Experiment 2. A positive correlation was found, r(52) = 0.83, p < 0.001. C) Test-retest reliability of Type 2 AUROC in Experiment 1. Participants' Type 2 AUROC in the first half of the study was significantly correlated with that in the second half of the study, r(51) = 0.43, p = 0.001. D) Test-retest reliability of Type 2 AUROC in Experiment 2. Again, a positive correlation was found between the two halves of the experiment, r(52) = 0.37, p = 0.006. All correlation coefficients were computed by Spearman rank tests. Lines of best fit were computed by the ordinary least squares method.

In addition, we calculated the split half reliability based on participants' control levels on the 8th to 13th reversal trials. High reliability was observed in both experiments (Experiment 1: r = 0.99, reliability = 0.997; Experiment 2: r = 0.97, reliability = 0.99) [38]. Similar results were obtained if the test was performed on control levels on the 7th to 13th or 6th to 13th reversal trials (Experiment 1, reversals 7–13: r = 0.995, reliability = 0.998; Experiment 1, reversals 6–13: r = 0.99, reliability = 0.997; Experiment 2, reversals 7–13: r = 0.97, reliability = 0.99; Experiment 2, reversals 6–13: r = 0.97, reliability = 0.99). To further validate the test-retest reliability of control threshold estimation, we also found that the two control thresholds estimated for each participant from the eighth through thirteenth reversals of each individual staircase were significantly correlated [Experiment 1: r(51) = 0.82, p < 0.0001; Experiment 2: r(52) = 0.81, p < 0.0001].

To again avoid the influence of early inflation of Type 1 performance, data after the third reversal trial was used to check the test-retest reliability of Type 2 AUROC estimates. A positive correlation was found between participants' Type 2 AUROC in the first and second half of

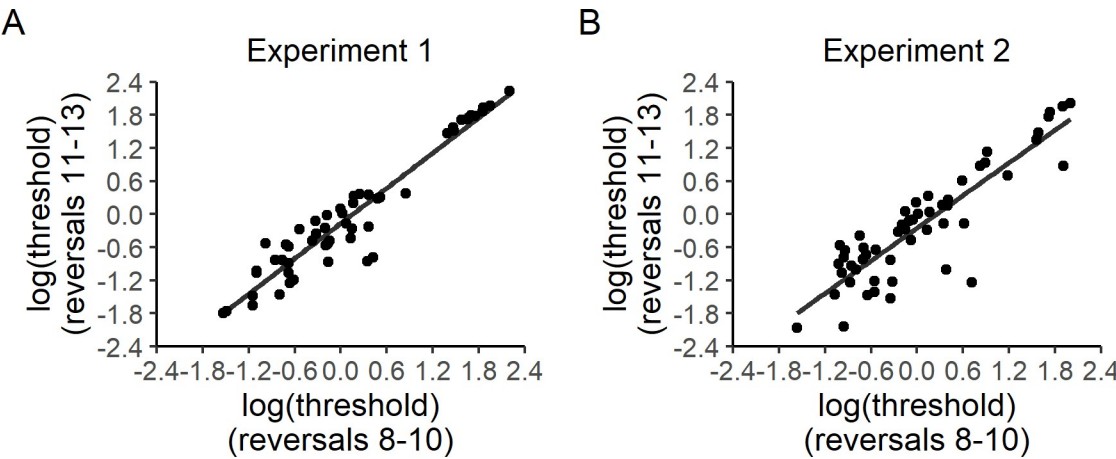

**Fig 4. Test-retest reliability of log-transformed control level thresholds in Experiments 1 (A) and 2 (B).** Spearman rank correlation coefficients for log-transformed control thresholds are the same as those reported for non-transformed control thresholds (Fig 3A and 3B). The transformed data shown here are intended to provide additional visual evidence that the observed test-retest reliability of control thresholds is not simply driven by extreme values.

both experiments [Experiment 1: r(51) = 0.43, p = 0.001, Fig 3C; Experiment 2: r(52) = 0.37, p = 0.006, Fig 3D].

## Discussion

The dot trajectory task introduced here was designed to quantify participants' sense of movement agency quickly and reliably. The present test-retest reliability results suggest that both agency control thresholds and metacognitive sensitivity can be reliably estimated from this relatively short (fewer than 100 trials) online thresholding procedure.

The current task offers several advantages over previous paradigms for quantifying self-agency. It avoids biases and memory confounds that may be inherent to paradigms using discrete timing judgments like the Libet clock paradigm [16] or those based on the intentional binding effect [3,39]. Further, it extends the benefits of paradigms that use interactive participant movements [30,31,40,41] in at least three ways. First, because the task is online, it can be easily implemented either at home or in laboratory or clinical settings. Second, the use of adaptive staircasing increases both flexibility and efficiency in the estimation of individual agency measures [42]. Third, the combined use of adaptive staircasing and confidence judgments allows for between-subjects comparisons of metacognitive sensitivity that are not confounded by Type 1 performance [43]. Importantly, this ensures that any potentially observable between-subjects differences in Type 1 control thresholds will not obscure any potential further between-subjects differences in metacognitive sensitivity. In this regard, future results from this task could potentially shed light on the extent to which the sense of agency is metacognitive in nature [44]. Finally, the significant test-retest reliability for agency control thresholds and metacognitive sensitivity observed in both experiments suggests that the task overcomes a common limitation of weak measurement stability in behavioral tasks [11]. However, to further confirm this, the current results should be replicated in a larger sample, as larger sample sizes can reduce estimates of measurement reliability [11].

Going forward, the next step will be to compare these measures between patients and healthy controls. Continuing with the example of FMDs, we hypothesize that patients' control level thresholds will be higher than that of healthy controls. The dot trajectory task requires participants to compare predicted dot trajectories with real trajectories. While the real

trajectories are represented by visual input, predicted trajectories are a combination of per-ceived trajectories and participants' inner prediction based on hand movements. Because FMD patients have impaired inner predictions [20], their predicted trajectories should be less accurate than those of controls. Therefore, patients with FMDs should require greater control levels to reach 75% accuracy on the dot trajectory task than controls. Similarly, because FMD patients display impaired somatosensory metacognition [21,26], we predict that they will have lower Type 2 AUROC compared to healthy controls.

The dot trajectory task may also have broader applicability for the diagnosis of other agency-related disorders whose primary symptoms are not movement-based. For example, some schizophrenia patients have demonstrated an excessive sense of self-agency, such that they tend to consider movements generated by others as being self-generated [41,45]. Such patients may experience difficulty distinguishing dot movements generated by themselves from those created by the program. This would be expected to manifest in a response profile similar to that predicted for FMD patients: higher Type 1 error rates (and thus, higher control thresholds) and inflated confidence ratings on incorrect trials (and thus, lower type 2 AUROC) relative to healthy participants.

The present study has a few limitations. First, participants provided Type 1 and Type 2 responses in two separate questions with a fixed order. This may allow information accumu-lated after the Type 1 response to be incorporated into Type 2 judgments, which could lead to inaccurate assessment of metacognitive sensitivity [46]. Thus, future studies may benefit from asking participants to indicate the target dot and rate confidence simultaneously [47]. Second, the lack of cursor position recording in the current study limited our ability to assess the influ-ence of participants' movement characteristics (e.g., velocity, shape of trajectories) on their mea-surements of self-agency. Therefore, experiment code that records mouse movements is provided in the online repository. This will allow future studies to explore the questions above. Third, since data was collected from each participant only once, we were not able to verify the task's test-retest reliability within participants and across days. Therefore, future studies should further assess test-retest reliability by collecting data across sessions separated by multiple days.

Lastly, the high exclusion rate in the present study may raise concerns about the effectiveness of the task; a task that yields usable data only 34.72% of the time would not seem to have much clinical utility. However, this concern is likely unfounded for a few reasons. First, high exclusion rates due to a lack of attentiveness to the task among mTurk participant samples have been reported previously [48,49, but see also 37]. Indeed, the significantly higher control thresholds observed among excluded participants putatively indicates a lack of engagement among this group. Thus, administering the task to non-mTurk participants, such as college students, may lower exclusion rates. Moreover, common methods of increasing participants' motivation, such as providing higher monetary rewards or gamifying the task could also be applied in future experiments. Importantly, we also expect that clinical patients will be less likely to have issues with reduced motivation or engagement, as the task is directly related to their condition.

In conclusion, the dot trajectory task introduced here is able to estimate two new measures of movement self-agency efficiently and reliably in healthy participants. We hope that future implementations of this novel task in studies with patients can improve our ability to both understand and diagnose agency-related disorders.

## Supporting information

**S1 Video. A video of the stimuli and task procedure.** An illustration of example trials with mouse movements side-by-side with the dot motion stimuli.
(MP4)

## Acknowledgments

We would like to thank Mouslim Cherkaoui and Angela Clague for helping conduct the study and providing useful feedback.

## Author Contributions

**Conceptualization:** Shiyun Wang, Hakwan Lau, J. D. Knotts.

**Formal analysis:** Shiyun Wang.

**Investigation:** Shiyun Wang.

**Methodology:** Shiyun Wang, Sivananda Rajananda, Hakwan Lau, J. D. Knotts.

**Project administration:** Hakwan Lau, J. D. Knotts.

**Software:** Shiyun Wang, Sivananda Rajananda.

**Visualization:** Shiyun Wang, J. D. Knotts.

**Writing – original draft:** Shiyun Wang, J. D. Knotts.

**Writing – review & editing:** Sivananda Rajananda, Hakwan Lau.

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
