## [Decision Letter · Decision Letter 0]

28 Oct 2020

PONE-D-20-24359

New measures of agency from an adaptive sensorimotor task

PLOS ONE

Dear Dr. Wang,

Thank you for submitting your manuscript to PLOS ONE. After careful consideration, we feel that it has merit but does not fully meet PLOS ONE’s publication criteria as it currently stands. Therefore, we invite you to submit a revised version of the manuscript that addresses the points raised during the review process.

The manuscript describes a very interesting task which could be of practical use, and I think it will be of interest to PLoS readers. However, both reviewers have a few concerns which need addressing before the manuscript can be published. In particular, Reviewer 1 has concerns about the utility of the task considering almost half of the participants had to be excluded. The rest of the concerns are fairly minor and should be easily addressed. 

We look forward to receiving your revised manuscript.

Kind regards,

Deborah Apthorp, Ph.D

Academic Editor

PLOS ONE

Additional Editor Comments:

Please pay careful attention to the reviewer's comments and revise your manuscript accordingly. In particular, Reviewer 1 expresses concern that almost half the participants in each experiment were excluded. Could the authors please respond to this concern?

Reviewers' comments:

Reviewer's Responses to Questions

**Comments to the Author**

1. Is the manuscript technically sound, and do the data support the conclusions?

Reviewer #1: Partly

Reviewer #2: Yes

2. Has the statistical analysis been performed appropriately and rigorously? 

Reviewer #1: Yes

Reviewer #2: Yes

3. Have the authors made all data underlying the findings in their manuscript fully available?

Reviewer #1: Yes

Reviewer #2: Yes

4. Is the manuscript presented in an intelligible fashion and written in standard English?

Reviewer #1: Yes

Reviewer #2: Yes

5. Review Comments to the Author

Reviewer #1: This study reports the results of an online motion tracking task to quantify movement agency. The procedures seem robust, although more details would be helpful.

Major

It is very concerning that almost half the participants in each experiment had to be excluded. A test that yields usable data only half the time does not seem like it would have much clinical utility. Why would so many people miss more than 40% of the catch trials? How long does the test take, on average? I don’t think that was mentioned in the manuscript. Is it long enough that many participants get bored? Do the authors have plans to make the test more engaging or other changes that might reduce the number of exclusions? These issues should at least be discussed.

More details about the task are needed. Did the two dots perform identical movements, apart from the cursor movement? Or did each dot have its own random motion? How were participants instructed to move the cursor? What if they don’t bother moving the cursor and just guess the answers? Were such trials excluded? Did some participants move the cursor more than others, or did the amount of cursor movement change over the course of the test?

Minor

Line 46. More details about the Libet clock paradigm would be helpful. How it works, what the patient is asked to do, etc.

Line 57-58. I’m not sure what this means. For readers who have no background knowledge of FMD, like me, it would be helpful to have more explanation. What effect does this mismatch have on the patient’s ability to function? How does it affect their daily lives?

The method of assessing test-retest reliability is a little unusual. Are there any plans to have participants complete the task two separate times, at least several days apart, and assess how consistent the results are within participants, across days?

Reviewer #2: In this study, Wang and colleagues introduce a new method for taking quantitative measurements of perceived agency, and demonstrate the effectiveness (and reliability) of the method in two online experiments with human subjects. Specifically, in their task two visual cursors are presented as subjects freely make movements. Critically, one cursor is yolked to subjects movements (under volitional control, with a titrated "control level") and the other moves randomly. Using a 2AFC and confidence ratings, the authors could measure the acuity of agency judgements under varying levels of visuomotor control. This technique is poised to improve these measurements relative to other techniques that suffer from various confounds, allowing for both better diagnostic capabilities and basic research into agency and perceived control.

This technique and study was very thorough, the methods were clear, and the test-retest reliability suggests that it is a very effective trait measurement. Overall, I recommend it for publication as it seems like a clear improvement on previous methods. I have several modest concerns/suggestions:

1) The paper could give a better sense of the observed movement data – what were subjects' movement velocities? What did their trajectories look like? Did they explore the whole workspace? Was there a relationship between specific kinematic properties (e.g. average velocity) and agency thresholds?

2) Figure 1A is not particularly illuminating – I would suggest showing several frames of the dots moving, perhaps with example movements of the subject below it, to really illustrate the task dynamics. Otherwise it's hard to get a sense of the phenomenological experience of the subjects, which is key in studies of high-level cognitive inferences like agency.

3) For the test-retest reliability correlations, why were the particular trial bins chosen (8-10/11-13)? These should be justified in the text somewhere.

Minor:

- misspelled reference to "Dannett" (ref 14) I believe should be "Dennett."

- it could be nice in the correlation figurers (i.e. fig 3) to have rho values depicted on the figure panel.

6. PLOS authors have the option to publish the peer review history of their article (what does this mean?). If published, this will include your full peer review and any attached files.

Reviewer #1: No

Reviewer #2: No

---

## [Author Response · Author response to Decision Letter 0]

17 Nov 2020

Please see the attached Response to Reviewers letter.

---

## [Decision Letter · Decision Letter 1]

3 Dec 2020

New measures of agency from an adaptive sensorimotor task

PONE-D-20-24359R1

Dear Dr. Wang,

We’re pleased to inform you that your manuscript has been judged scientifically suitable for publication and will be formally accepted for publication once it meets all outstanding technical requirements.

Kind regards,

Deborah Apthorp, Ph.D

Academic Editor

PLOS ONE

Additional Editor Comments (optional):

Reviewers' comments:

Reviewer's Responses to Questions

**Comments to the Author**

1. If the authors have adequately addressed your comments raised in a previous round of review and you feel that this manuscript is now acceptable for publication, you may indicate that here to bypass the “Comments to the Author” section, enter your conflict of interest statement in the “Confidential to Editor” section, and submit your "Accept" recommendation.

Reviewer #1: All comments have been addressed

Reviewer #2: All comments have been addressed

2. Is the manuscript technically sound, and do the data support the conclusions?

Reviewer #1: Yes

Reviewer #2: Yes

3. Has the statistical analysis been performed appropriately and rigorously? 

Reviewer #1: Yes

Reviewer #2: Yes

4. Have the authors made all data underlying the findings in their manuscript fully available?

Reviewer #1: Yes

Reviewer #2: Yes

5. Is the manuscript presented in an intelligible fashion and written in standard English?

Reviewer #1: Yes

Reviewer #2: Yes

6. Review Comments to the Author

Reviewer #1: (No Response)

Reviewer #2: my comments have been addressed.

my only further suggestion for later versions of the task itself is to continuously record and save mouse positions to afford more in-depth analysis of movements.

7. PLOS authors have the option to publish the peer review history of their article (what does this mean?). If published, this will include your full peer review and any attached files.

Reviewer #1: No

Reviewer #2: No

---

## [Editor Report · Acceptance letter]

7 Dec 2020

PONE-D-20-24359R1 

New measures of agency from an adaptive sensorimotor task 

Dear Dr. Wang:

I'm pleased to inform you that your manuscript has been deemed suitable for publication in PLOS ONE. Congratulations! Your manuscript is now with our production department. 

Kind regards, 

on behalf of

Dr. Deborah Apthorp 

Academic Editor

PLOS ONE